# Microcontroller-Optimized Measurement Electronics for Coherent Control Applications of NV Centers

**DOI:** 10.3390/s24103138

**Published:** 2024-05-15

**Authors:** Dennis Stiegekötter, Jens Pogorzelski, Ludwig Horsthemke, Frederik Hoffmann, Markus Gregor, Peter Glösekötter

**Affiliations:** 1Department of Electrical Engineering and Computer Science, FH Münster—University of Applied Sciences, Stegerwaldstraße 39, 48565 Steinfurt, Germanyl.horsthemke@fh-muenster.de (L.H.);; 2Department of Engineering Physics, FH Münster—University of Applied Sciences, Stegerwaldstraße 39, 48565 Steinfurt, Germany

**Keywords:** NV center in diamond, quantum sensing, microcontroller, Rabi oscillation, ODMR, handheld, *π* pulse

## Abstract

Long coherence times at room temperature make the NV center a promising candidate for quantum sensors and quantum computers. The necessary coherent control of the electron spin triplet in the ground state requires microwave π pulses in the nanosecond range, obtained from the Rabi oscillation of the m_S_ spin states of the magnetic resonances of the NV centers. Laboratory equipment has a high temporal resolution for these measurements but is expensive and, therefore, uninteresting for fields such as education. In this work, we present measurement electronics for NV centers that are optimized for microcontrollers. It is shown that the Rabi frequency is linear to the output of the digital-to-analog converter (DAC) and is used to adapt the time length π of the electron spin flip, to the limited pulse width resolution of the microcontroller. This was achieved by breaking down the most relevant functions of conventional laboratory devices and replacing them with commercially available integrated components. The result is a cost-effective handheld setup for coherent control applications of NV centers.

## 1. Introduction

The nitrogen-vacancy center is a solid-state emitter that emits photons by optical pumping with a dependency on the photon intensity from the internal electron spin. This property makes the NV center interesting for an application as a magnetic sensor and is being investigated in detail in this respect. Some integrated sensors have already demonstrated that they could achieve sensitivities of 344 pT/$\sqrt{\mathrm{Hz}}$ and that the performance can already exceed those of conventional Hall sensors [1]. The focus is often on miniaturizing the sensor head with the diamond, radio frequency (RF) excitation via an antenna, and a photodiode for detection as a compact component [1,2,3,4]. However, these sensors still require laboratory measuring instruments to evaluate the weak photocurrent signal and are, therefore, anything but easy to transport. Thus, the main focus is on the electronics behind it and not on the integration of a sensor head. The electronics include a switchable source for microwaves in the GHz range, a detector for photodiodes, a lock-in amplifier (LIA) with a low-pass filter for signal acquisition, a microcontroller for the output of pulse sequences, and the data acquisition from the LIA and the control of the microwave source. Research in this direction has been carried out already but does not show a completely integrated solution and is either limited to FPGA-based development boards or the simple plugging together of COTS modules [5,6,7]. Pulse sequences needed for various spin manipulation experiments can be programmed using a simple array and can be modified on the fly. The central control of the lock-in reference together with the laser driver and the microwave guarantees synchronization of the individual ICs, which makes phase correction with the LIA obsolete. These features—given by the microcontroller as the central control unit for TIA, LIA, laser drivers, and microwaves—considerably simplify handling. The structure is designed for a ±9 V power supply, allowing operation from block batteries. The result is a portable system that can be carried in a pocket. To demonstrate the function, measurements of ODMR and Rabi oscillation are presented, and the π/2 time can be adjusted to the output pulse length of the microcontroller with little effort via DAC feedback. This eliminates the need for a high temporal resolution of the pulse sequences, which is otherwise necessary to reduce errors in coherent control.

## 2. Background

The NV center defect in the diamond crystal structure consists of a substituted nitrogen atom next to a vacancy in the carbon lattice and can be oriented in a total of four spatial orientations. The NV center can be described in a simplified view by an energy level diagram of the spin-triplet states (^3^A and ^3^E) as shown in Figure 1a for a single NV center, which is subdivided into the m_S_ spin states, and the spin singlet states (^1^A and ^1^E) of the electrons. The energetic transition from the ground state ^3^A to the excited state ^3^E can be achieved by off-resonant excitation with a laser at a wavelength of 520 nm. By decaying back to the ground state ^3^A, the NV center emits photons in a wavelength range between 637 nm and 800 nm. In the excited state ^3^E, there is also the possibility of decaying back to the ground state ^3^A via the singlet states ^1^A, ^1^E. The probability of this decay path is higher for electrons in the $m_{S}=\pm 1$ state than for electrons in the $m_{S}=0$ state. This means that electrons that cycle between the $m_{S}=\pm 1$ ground state and the $m_{S}=\pm 1$ excited state emit fewer photons than electrons that cycle between the $m_{S}=0$ ground state and the $m_{S}=0$ excited state. The photon difference between $m_{S}=0$ state and $m_{S}=\pm 1$ leads to an optical contrast of the luminescence of a maximum of 30% [8,9,10]. The cycle of optical excitation to the ^3^E state followed by decay to the ground state ^3^A is spin-preserving. However, if the decay occurs via the singlet state (^1^A, ^1^E) to the ground state ^3^A, a transition to the $m_{S}=0$ state is possible. This initialization results in a lift of the thermal equilibrium between the $m_{S}=0$ and $m_{S}=\pm 1$ states. The spins can be flipped, while still being in the initialized state, between the $m_{S}=0$ and $m_{S}=\pm 1$ states by an oscillating magnetic field, $B_{\mathrm{mw}}$, applied perpendicular to the NV axis. This field’s frequency must resonate with the Larmor frequency of the electrons. The zero-field splitting (ZFS) of the $m_{S}=\pm 1$ states is at D=2.87 GHz and is split apart by an external magnetic field $B_{||\mathrm{NV}}$ in parallel alignment to the NV axis with the gyromagnetic ratio γ=28 GHz/T by $2\gamma B_{||\mathrm{NV}}$.

### 2.1. Measurement Protocol

The theoretical 30% contrast is not achieved in the application. This value refers to a single NV center. However, the measurements were performed on a micro-diamond containing an ensemble of NV centers. The lattice structure of the diamond allows the NV centers to adopt four different crystal directions, resulting in the ensemble being equally distributed across these directions. The major factor leading to reduced contrast is the background fluorescence of the NV centers that are not excited to the $m_{S}=\pm 1$ state via microwave excitation because the frequency of the microwave can only be tuned to one resonance [8,11]. This causes the SNR to deteriorate. A LIA and a boxcar averager (BCA) are, therefore, implemented in the hardware to measure the weak fluorescence change. The LIA multiplies input signals by a specified reference frequency and filters the mixed signals with a low-pass filter. The result is a bandpass characteristic that detects signals within the low-pass filter bandwidth, centered around the reference frequency. For the reference frequency $f_{ref}=1/T$ as a symmetrical square wave and a low pass filter at the output, with a time constant τ=RC>>T=2π/ω, the procedure is simplified to a difference of the mean values in both reference half-waves [12].
(1)$$\langle U_{\mathrm{sig}}\rangle =\frac{2}{T}\int_{0}^{\frac{\pi }{\omega }}U_{\mathrm{PLsig}}\left(\omega t\right)\;dt-\frac{2}{T}\int_{\frac{\pi }{\omega }}^{\frac{2\pi }{\omega }}U_{\mathrm{PLsig}}\left(\omega t\right)\;dt.$$
In order to measure the spin signals from the fluorescence of the NV centers, these spin signals must be modulated at the reference frequency. Figure 2a illustrates the principle. During the first reference half-wave, a laser pulse polarizes the spins to the $m_{S}=0$ state, followed by a microwave pulse that manipulates the spins. A second laser pulse reads out the diamond and simultaneously repolarizes the spins back to the $m_{S}=0$ state. In the second reference half-wave, the microwave remains switched off. This leads to a difference in the mean values between spin-manipulated fluorescence in the first reference half-wave and non-spin-manipulated fluorescence in the second reference half-wave [8]. However, signals with a very low duty cycle, such as the $T_{1}$ measurement, lead to a deterioration in the signal-to-noise ratio (SNR), making the LIA no longer suitable.

In this case, a BCA is the better choice. A BCA multiplies the input signal by a rectangular pulse train, as shown in Figure 2b, which means that only signal components within the boxcar window are integrated. Noise that lies between the pulses is ignored. The integrated signal is averaged over *N* periods. The BCA measures the change in the fluorescence area of the diamond and can use the same pulse sequences as the LIA. A laser pulse polarizes the spins to the $m_{S}=0$ state and a subsequent microwave pulse manipulates the spins (or, as in the case of the $T_{1}$ measurement, without microwaves at all). A second laser pulse then reads the NV centers and the fluorescence is integrated. The process is repeated N times and the mean value of the area is calculated.

### 2.2. Detection of Spin Signals

A simple permanent magnet, generating the magnetic bias field $B_{0}=15$ mT, is placed near the diamond. The four different orientations of the NV center axis result in eight detectable magnetic resonances. Based on the ODMR pulse pattern outlined in Figure 2a, the measurement is repeated in 1 MHz steps with a pulse width of $\tau_{\mathrm{mw}}$ = 1.6 μs, at a power of 17 dBm. Feeding ten repetitions of the ODMR pattern for the laser and microwave into the low-pass filter in the first half-wave of the LIA reference [8] results in the fluorescence signal in Figure 1b. The Lorentz-shaped signals in the frequency range from 2.55 GHz to 3.25 GHz correspond to the contrast differences of an NV axis as soon as the frequency is in resonance with the transitions $m_{S}=0$ and $m_{S}=\pm 1$. The eight different resonant frequencies result from different $B_{||\mathrm{NV}}$ for each axis as a result of the different orientations in the diamond and their angle to the $B_{0}$ field of the permanent magnet. The low contrast of axis 1 is caused by its orientation in the diamond (more about the axis orientation in Section 4). The decrease in contrast toward low and high frequencies is due to the 400 MHz bandwidth of the antenna used. Deviating center frequencies (CF) from the ZFS with CF1 = 2.934 GHz, CF2 = 2.933 GHz, CF3 = 2.897 GHz, and CF4 = 2.887 GHz are calculated from resonances 1–4. The drift is due to the non-axial magnetic field component $B_{⊥}$, which is present at the NV center and cannot be neglected for magnetic flux densities in the 10 mT range [13].

A simple method to determine the $T_{1}$ relaxation is to polarize the spins according to the pulse pattern in Figure 2b and optically read out the distribution of the $m_{S}$ states after time $\tau_{\mathrm{spacing}}$. Figure 1c shows the calculated fluorescence area as an exponentially decaying curve as a function of the pulse spacing $\tau_{\mathrm{spacing}}$ in 100 μs steps, for a time interval of 5 ms. This measurement is performed without a microwave and only the laser is pulsed. In the time $\tau_{\mathrm{spacing}}$ between the laser pulses, the polarized spins decay from the $m_{S}=0$ state and reoccupy the $m_{S}=\pm 1$ states. With increasing $\tau_{\mathrm{spacing}}$, the occupation also increases until thermal equilibrium is reached again. The proportion of spins in the $m_{S}=\pm 1$ states increases the contrast and leads to a falling curve in the output signal. The measured $T_{1}$ time is 0.855 ms and is, thus, within the expected time range of NV centers in HPHT diamonds, which have a relaxation time constant of a few milliseconds at room temperature [14].

For more complex measurements, such as the $T_{2}$ time constant, coherent control of the electron spin is required. Coherent control is achieved when the spins can be controlled from the initial state $m_{S}=0$ to a target state by an external alternating magnetic field $B_{\mathrm{mw}}$ perpendicular to the NV axis. The time to flip the electron spin from the $m_{S}=0$ state directly to the $m_{S}=\pm 1$ state is called π pulse and is determined in Section 4 with a time of 106.6 ns by measuring the Rabi oscillation.

$T_{2}$ describes the transverse decay, which results from a change in the phase coherence of the precessing spins. Since the phase coherence of the spins drifts apart at a constant rate in the case of static inhomogeneities, this process can be reversed by a π pulse, so that the phases of the spins drift toward each other at the same rate. This approach is also known as the Hahn echo sequence [15], outlined in Figure 2a. The $T_{2}$ measurement in Figure 1d examines the single resonance at 2.878 GHz with a power $P_{\mathrm{mw}}=17$ dBm. First, the spins are polarized to the $m_{S}=0$ state and brought into superposition between $m_{S}=0$ and $m_{S}=-1$ state by a π/2 = 53.3 ns microwave pulse. The spins then ‘precess’ freely for a time τ and are varied in a range between 106 ns to 5.2 μs with a step size of 53.3 ns. A π pulse then changes the direction of propagation of the phases for a further τ before readout. The exponential curve of the LIA signal is caused by interactions between the electron spins with the spin bath within the time τ of free precession. The influence of the spin bath can be described as fluctuating magnetic fields at the NV center [16] that arise in the diamond mainly from impurities with P1 centers [17,18]. This leads to a dephasing of the electron spins in superposition. At the end of the sequence, some spins are, therefore, in the $m_{S}=-1$ state again. The occupation of the $m_{S}=-1$ state increases with increasing τ and leads to a rise in contrast. $T_{2}^{\mathrm{Hahn}}$ = 674 ns was measured from the output signal. A low $T_{2}$ time in the lower μs or ns range indicates high contamination of the diamond with P1 centers [17] and determines the time range in which coherent control can be applied before the signal is lost in noise.

The measurements in Figure 1b–d confirm the function of the measuring device. However, the measurements also show that the performance is dependent on the applied static magnetic field $B_{0}$. A larger $B_{0}$ field splits the resonances further outside of the bandwidth of the antenna and the contrast of the resonances reduces significantly. The larger splitting also requires a wider frequency range to be scanned in order to capture all resonances. Due to the fixed time constant of the LIA filter, the scanning time for the measurement increases if the frequency step size is kept constant. For the application as a sensor, it must be taken into account that the resonances do not evolve linearly with the magnetic field because of the non-axial magnetic field component $B_{⊥}$.

## 3. Hardware Development

Both the detection and the excitation of the spin-dependent fluorescence are performed via the modules shown in Figure 3 and their interconnection. The pulse sequences for the microwave generator and laser driver required for spin manipulation are controlled centrally by a microcontroller (STM32G491, STMicroelectronics, Geneva, Switzerland). The photodiode signals (S5971, Hamamatsu Photonics K.K., Hamamatsu City, Japan) are processed by an analog LIA (AD630, Analog Devices Inc., Norwood, MA, USA) with a following low-pass filter at a cut-off frequency of 1 Hz. The sample rate of the analog-to-digital converter (ADC) is reduced to the settling time of the filter. The data are transmitted via a universal serial bus interface and can be plotted on a PC. To excite the resonances, a serial peripheral interface (SPI) controlled phase-locked loop (PLL) (ADF4351, Analog Devices Inc., Norwood, MA, USA) with an output frequency range between 35 MHz and 4400 MHz is used. The amplitude can be adjusted with the DAC 1 signal of the microcontroller via an I/Q modulator (LTC5589, Analog Devices Inc., Norwood, MA, USA) to vary the microwave power.

### 3.1. Bootstrapped Transimpedance Amplifier

The photodiode is mounted at the end of a shielded cable as shown in Figure 4a and is connected to the transimpedance amplifier (TIA) with a sub-miniature version A (SMA) screw connector so that the measuring electronics have a purely electrical interface and do not require any optical components such as fiber couplers, dichroic mirrors, etc. A high SNR is crucial for detecting weak signals with sufficiently high bandwidth to follow the edges of the pulses. In conventional NV readout methods, a difference in the fluorescence rate between the $m_{S}=0$ state and the $m_{S}=\pm 1$ state can be measured in the first 500 ns [18,19]. The bandwidth of the detector should therefore be at least 2 MHz. The incoming photons of the NV centers are Poisson-distributed and are detected as individual particles in the PN junction of the photodiode. This process is stochastic and leads to a noise current $i_{\mathrm{Nshot}}=\sqrt{2qI}$ where *q* is the elementary charge and *I* is the photocurrent. It is, therefore, independent of the temperature and increases with the root of the photocurrent. Thermal noise, on the other hand, is caused by fluctuations in the charge carrier velocity in the conduction band of the TIA feedback resistor $R_{f}$. These fluctuations are caused by thermal energy and lead to a noise current $i_{\mathrm{Nth}}=\sqrt{4kT/R}$ with the Boltzmann constant *k*. The noise current is in turn independent of other currents and depends on the temperature and the resistance value. In optical measurements, the shot noise sets the limit for the sensitivity. Improvements in the SNR can then only be achieved by increasing the photocurrent or lowering the bandwidth. Therefore, the construction of the TIA should ensure that the shot noise dominates within the desired bandwidth at a given photocurrent. The SNR of the signal depends on the ratio of the two noise sources $i_{\mathrm{Nth}}/i_{\mathrm{Nshot}}$, as well as the input current noise $i_{\mathrm{Namp}}$ and input voltage noise $e_{n}$ of the operational amplifier (OpAmp). The SNR decreases if one of these noise sources becomes larger compared to the shot noise [20]. A simple way to reduce $i_{\mathrm{Nth}}$ is to increase the resistance value $R_{f}$. However, a larger $R_{f}$ also reduces the bandwidth of the detector and forms, with the input capacitance of OpAmp and the capacitances of the photodiode as well as the line, a low-pass filter with the cutoff frequency $f_{\mathrm{RC}}=1/2\pi R_{f}C_{\mathrm{in}}$. An acceptable compromise is achieved with $i_{\mathrm{Nth}}=0.5\cdot i_{\mathrm{Nshot}}$, which is accompanied by a loss of 1 dB in SNR and an output voltage at the TIA of 200 mV at room temperature [20]. $i_{\mathrm{Namp}}$ and $e_{n}$ depend on the specification of the OpAmp. For FET inputs, $i_{\mathrm{Namp}}$ is small compared to $e_{n}$ and the input voltage noise dominates for low light intensity with large $R_{f}C_{\mathrm{in}}$. An equivalent noise current $i_{N}=e_{n}\omega C_{\mathrm{in}}$ results from $R_{f}$ and $C_{\mathrm{in}}$, which is proportionally dependent on the input capacitance and the angular frequency ω [12,20]. To increase the bandwidth and minimize the influence of $i_{N}$, the capacitive load at the summing node of the OpAmp can be reduced with a bootstrap circuit. So, when choosing a suitable amplifier, $e_{n}$ should be as small as possible. Also, when selecting a suitable photodiode, care should be taken to ensure a low junction capacitance, $C_{d}$. The value depends on the expansion of the space charge region and can be decreased with little effort by a negative bias at the anode by a factor of 7–10, this only increases the leakage current and decreases the SNR for negligible 0.004 dB [12,20]. Since the photodiode is at the end of a shielded cable, a non-negligible capacitance, $C_{l}$, is also introduced with ≈100 pF/m [12]. To counteract this, an AC-coupled bootstrap is connected between the summing node of the TIA and the cable. Figure 4a shows the entire circuit. The bootstrap copies a replica of the high-side signal, AC coupled to the low-side of the cable output. The replicated signal reduces recharging of the effective capacitances of the photodiode and cable, which decreases the effective capacitance at the summing node and attenuates it by typically a factor of 10. A signal at the summing node is copied across the junction field effect transistor (JFET) follower $Q_{1}$ and AC coupled through a buffer $Q_{2}$ via capacitor $C_{\mathrm{ac}}$ to the anode of the photodiode. Negative bias is applied across $R_{1}$ and prevents shorting with the signal across $C_{\mathrm{ac}}$ [12]. $R_{2}$ and $R_{4}$ are dimensioned so that a voltage of 0 V is applied to the buffer output at rest. $R_{3}$ limits the current through JFET $Q_{1}$ with $R_{4}$. The feedback capacitance $C_{f}$ is calculated from
(2)$$C_{f}=\frac{1}{2\pi R_{f}\sqrt{f_{T}\cdot f_{RC_{\text{in,eff}}}}}$$
with $f_{T}$ as the transit frequency of the used OpAmp and $C_{\mathrm{in},\mathrm{eff}}=\frac{1}{10}\left(C_{d}+C_{l}\right)+C_{\mathrm{rss}}+C_{\mathrm{amp}}$ effective capacitance. $C_{d}$ is the capacitance of the photodiode, $C_{l}$ is the capacitance of the cable, $C_{\mathrm{rss}}$ is the reverse transfer capacitance, and $C_{\mathrm{amp}}$ is the capacitance of the OpAmp input. For exact component values see Section A.1.

### 3.2. Lock-In-Amplifier and Boxcar Averager

The luminescence signals from the NV centers provided by the TIA are demodulated in the next step using the analog LIA. The demodulator is specified with a bandwidth of 350 kHz, a slew rate of 45 V/μs, and a signal amplification of 100 dB. The input signal is mixed with a square wave by alternately switching the positive amplifier (AMP) A and the negative AMP B to the output filter by means of a reference frequency-controlled FET switch. The internal structure is sketched in Figure 4b with the corresponding component values in Section A.2. AMP A closes the path to the filter in the first half of the reference signal, while AMP B closes the path to the filter in the second half of the reference signal. To allow the amplitude response to fall sharply outside the 1 Hz range, the filter has an order of N=3 and thus an attenuation of 60 dB per decade. Figure 4b shows the filter cascade at the output of the LIA, consisting of a first-order inverting low-pass filter and a second-order multiple-feedback topology in series. Compared to the Sallen–Key design, the MFB filter has a higher quality factor and is better suited for larger gains [21]. The weak fluorescence signals are amplified by the filter with a factor of 100. The transfer function of the entire filter is calculated by multiplying the transfer functions of the individual filters to give
(3)$$G\left(s\right)=\frac{A_{1}}{1+a_{1}s}\cdot \frac{A_{2}}{1+a_{2}s+b_{2}s^{2}}$$
The filter coefficients determine the frequency response and are selected according to Table 1 with a Butterworth characteristic to provide a maximally flat passband [21].

A boxcar averager is implemented by using the PWM channel of timer 2 for pulse output to the laser driver. At the same time, the counter status is monitored in a second channel of the timer and an interrupt is triggered before the PWM output switches to a high level. The TIA is connected to a general-purpose input/output (GPIO) port to which two ADCs have access. The two ADCs achieve a sampling rate of 8 Msps with 12-bit resolution in interleaved mode and sample the GPIO port alternately. The samples of the ADCs are stored in a common data register. Within the interrupt routine, the processor reads the data register each time the end-of-conversion flag of each ADC is set. The sampling process is repeated until a total of ten register values have been recorded. The integral is then calculated from the recorded samples according to the extended Simpson’s rule [22]
(4)$$\int_{x_{1}}^{x_{N}}f\left(x\right)\;dx=\frac{h}{3}\left(\sum_{n=2}^{N-1}\left({\left(-1\right)}^{n}+3\right)x_{n}\right)+\frac{h}{3}\left(x_{1}+x_{N}\right)$$
with h=125 ns as the time difference between sampled values. Equation (Equation 4) is executed by the microcontroller after the interrupt routine and the calculation result is processed to determine the average value.

### 3.3. Microwave Source

The internal electron spin of the NV centers is manipulated by excitation with microwaves that resonate with the Larmor frequency of the electrons. These microwaves are generated using a voltage-controlled oscillator (VCO) and can be set to precise values within a controlled loop. The design of the filter and the output of the PLL correspond to the topologies from the data sheet of the ADF with a balun at the differential outputs $RF_{\mathrm{out}}A+$ and $RF_{\mathrm{out}}A-$ for 3 dB more output power compared to a single-ended application. An overview of the wired ICs can be seen in Section A.3. The PLL serves as a local oscillator for the I/Q modulator, its in-phase and quadrature inputs are differential at a common mode voltage level of 1.4 V and are driven by the microcontroller’s 12-bit digital-to-analog converter. The signal at frequency $f_{\mathrm{LO}}$ is fed into the I/Q modulator via an LC matching network, passing through a phase shifter, and is split into two paths that are 90° out of phase with each other. The modulator supports DC coupling at the inputs and can vary the output power with the DAC (with differential mode driver). The external circuit design is based on its evaluation board (DC2391A, Analog Devices Inc., Norwood, MA, USA). RF switches (M3SWA2, Mini-Circuits, Brooklyn, NY, USA) are used to pulse the microwave with a duration of $\tau_{\mathrm{mw}}$. These have a rise/fall time of 6 ns and are switched via transistor-transistor-logic (TTL) logic. As the output power of the PLL has a maximum of 5 dBm and is subject to further losses due to lossy transmission lines, I/Q modulator, and RF switches, the modulated and pulsed output frequency is amplified by a gain stage with three adjustable levels, as shown in Figure 5a. The antenna for microwave excitation corresponds to the design of Sasaki et al. [23]. They achieved a Rabi frequency of 4.6 MHz at a power $P_{\mathrm{mw}}=1$ W using an N-doped type IIA substrate diamond [23]. This indicates that due to the angle between $B_{\mathrm{mw}}$ and the NV axes, less power is required with an NV axis alignment to achieve the same Rabi frequency. Therefore, the first path (1) contains a cascade of two gain block amplifiers (GRF2012, Guerrilla RF Inc., Greensboro, NC, USA), which each have a gain of 14.8 dB and a 1-dB compression point of 23 dBm at a voltage supply of 5 V and a current of 100 mA. The resulting amplification of the cascade is therefore 29.7 dB. The second path (1.1) passes through a single amplifier with a gain of 14.8 dB. At the third path (1.2), the microwave is provided without gain via a coaxial jack and corresponds to the power of the I/Q modulator minus attenuations due to switches as well as line reflections. In any case, we can control the power by the DAC and I/Q modulator.

### 3.4. Microcontroller

The control of the individual modules on the board is handled by the microcontroller with an ARM Cortex-M4 processor at a clock frequency of 150 MHz. In addition to calculating the register contents of the I/Q modulator and the PLL, the pulse sequences are implemented using the GPIO peripherals. In order to be able to stream more complex pulse patterns such as the Hahn-echo-Sequence, functions like pulse–width–modulation are unsuitable. The simplest solution is to update the GPIO ports at equidistant time points and switch the corresponding ports on or off, as required. The information about which GPIO outputs are set or reset is stored in a 1 × 1440 32-bit array in the SRAM2 memory to modulate the laser and microwave and provide a reference for the LIA. Placement in the SRAM2 is necessary to ensure the lowest possible jitter so that the bus connection between the DMA controller and GPIO port A is free of all other signals and there is no transmission delay. These bit masks are written via the direct memory access (DMA) controller directly into the GPIO bit set and reset register (BSRR) in which the first half-word defines the pins to be set and the second half-word specifies which ones are reset. Pulses can be generated at the respective outputs by means of periodic write commands, the length of which depends on the period duration and the interval between the set and reset instructions. A DMA data transfer is triggered via the TRGO signal at each update event of timer 2 and increments the DMA pointer to the next position in the array. The entire process requires eight clock cycles, which results in the minimum pulse width of $T_{p,\min}=8/150\;\mathrm{MHz}=53.3$ ns. The size of the array determines the reference frequency with $1/L_{A}T_{p,\min}$. Once started, the array is passed through circularly. Every 700 ms an interrupt of timer 6 is triggered, via which the callback function takes a sample of the filter. Afterward, either the frequency for ODMR or the pulse pattern can be modified for the next sample.

## 4. Setup and Optimization

The NV diamond used is a high-pressure, high-temperature (HPHT) 150 μm micro-diamond (MDNV150umHi50mg, Adámas Nanotechnologies, Raleigh, NC, USA) with a NV concentration between ≈2.5–3 ppm. The sample is located in the hole of the planar ring antenna, with a spatially homogeneous magnetic field distribution within an area of 1 ${\mathrm{mm}}^{2}$ [23] parallel to one of the [111] NV axes. The orientation of the diamond is determined by its geometry. Diamonds grow in discrete directions with the crystal axes as the normal vector to the growth surface. Hexagonal sectors correspond to the NV axes, while octagonal sectors are assigned to the [100], [010], and [001] directions [24]. In Figure 5c, the hexagonal structure can be clearly recognized by the top view. To confirm the orientation, the angle to the (001) surface is measured at a 90° angle to the [111] axis as shown in Figure 5d. In theory, the calculated angle between the (111) and (001) surface is 125.3°, and the measured values are between 120° and 130° and, thus, lie in an area that excludes any confusion with another NV surface that is separated by 109.28°.

The printed circuit board (PCB) antenna is clamped in a 3D-printed structure (see Figure 5b) and is located vertically below a laser diode (PLT5 520B, OSRAM GmbH, Munich, Germany). The laser is encapsulated in an adjustable collimator (LDA-56, Roithner Lasertechnik GmbH, Vienna, Austria) and is emitting light at a wavelength of 520 nm. To collect the fluorescence, the photodiode is placed at an angle to the antenna with a glued-on long-pass filter (OG590, Schott AG, Mainz, Germany) and is connected via a shielded cable to suppress interfering signals. The largest contrast in Figure 1b results for the resonances with the crystal orientation [$11\bar{1}$] and [$\bar{1}\bar{1}1$] (see Figure 5c), which are approximately perpendicular to the applied bias magnetic field and lie within the 400 MHz antenna bandwidth. The innermost resonances correspond to the [111], [$\bar{1}\bar{1}\bar{1}$] NV axis and emit with low contrast, as they are aligned parallel to the alternating magnetic field $B_{\mathrm{mw}}$ of the antenna. The resonance splitting and crystal orientation enable a single resonance to be filtered out that deviates by 19.23° from the vertical of the antenna’s alternating magnetic field $B_{\mathrm{mw}}$. A good microwave excitation plays an important role in measuring the Rabi oscillation when it comes to obtaining a fast fundamental Rabi frequency. Therefore, the microwave frequency is fixed to the $m_{S}=-1$ resonance at 2.822 GHz and a sweep of the pulse width $\tau_{\mathrm{mw}}$ is run to measure the Rabi oscillation. The oscillation as a function of the pulse length $\tau_{\mathrm{mw}}$ can be described with an exponentially decaying sine function.
(5)$$S_{\mathrm{rabi}}\left(\tau_{\mathrm{mw}}\right)=A\sin\left(2\pi f_{\mathrm{rabi}}\cdot \tau_{\mathrm{mw}}+\phi \right)e^{-\frac{\tau_{\mathrm{mw}}}{T_{2}^{\mathrm{rabi}}}}+m\tau_{\mathrm{mw}}+b$$
where *A* is the amplitude of the Rabi oscillation, $f_{\mathrm{rabi}}$ is the Rabi frequency, ϕ is the phase offset, $T_{2}^{\mathrm{rabi}}$ is the decay time constant, *m* is the slope to counteract temperature drift and *b* is the signal offset [8]. The orange curve in Figure 6a depicts a π pulse at a microwave power of 17 dBm with approximately 106.6 ns.

### Control of Rabi Frequency

If the time for a π pulse does not match the time resolution of the microcontroller $T_{p,\min}$, or integer multiples $NT_{p,\min}$, the applied operations lead to coherent errors. To reduce these errors, it is not necessary to increase the resolution if the Rabi oscillation can be matched to any value for a π pulse through the control of the microwave power. Since $B_{\mathrm{mw}}\propto I_{\mathrm{mw}}=\sqrt{P_{\mathrm{mw}}}/R$, the Rabi frequency should be linearly dependent on the square root of the power. For verification, the output power is attenuated in 9 dB steps and contrasted in Figure 6a. These are fitted to an exponentially decaying sinusoidal function, and the respective Rabi frequency is read from the fit parameters. Then, the power $P_{\mathrm{mw}}$ is determined and the square root is taken. The comparison between $\sqrt{P_{\mathrm{mw}}}$ and $\Omega_{R}$ in Figure 6b shows the linear relationship. The I/Q modulators I and Q inputs are analog mixers whose input signal is multiplied by the local oscillator signal. For a DC voltage from the DAC to I, the amplitude of the output frequency becomes the product $U_{\mathrm{mod}}\left(t\right)=U_{\mathrm{dac}}U_{\mathrm{lo}}\left(t\right)$ and is proportional to the DAC voltage. This means that due to the relationship $U=R\sqrt{P}$, the square root of the output power is also in a linear relationship with $U_{\mathrm{dac}}$. The output frequency is fixed at 2.87 GHz in continuous wave (CW) mode, and the output power is recorded for all 100 DAC step values. The individual measurement points are converted to power ($P_{\mathrm{mw}}$) before the root is taken.

Figure 6c shows a clear, linear relationship between $\sqrt{P_{\mathrm{mw}}}$ and the DAC value. The measured data points are plotted in red, while the blue represents the fit using a linear magnitude function. The characteristic V-shape results from the hardware implementation. The I and Q inputs of the modulator are differentials around an average value of 1.4 V. If the DAC is increased, the differential voltage decreases and overlaps at the point $U_{\mathrm{dac}}/2$ before it increases again. The result is the characteristic V-shape. From the two results, we can now conclude that the frequency of the Rabi oscillation is linearly dependent on the voltage level of DAC. This observation makes it very useful to fit the π or π/2 time to the limited resolution of $T_{\mathrm{mw}}$. With π as the half oscillation period and $T_{\mathrm{mw}}=NT_{p,\min}$, we obtain $\Omega_{R}=2\pi /\left(2NT_{p,\min}\right)$, with which the respective DAC value over both straight-line equations is determined to
(6)$$\sqrt{P_{\mathrm{mw}}}=\frac{\Delta \sqrt{P_{\mathrm{mw}}}}{\Delta \Omega_{R}}\Omega_{R}+b_{\Omega_{R}}$$
(7)$$\sqrt{P_{\mathrm{mw}}}=\frac{\Delta \sqrt{P_{\mathrm{mw}}}}{\Delta C}C+b_{C}$$
(8)$$\Rightarrow C=\frac{b_{\Omega_{R}}-b_{C}}{\Delta \sqrt{P_{\mathrm{mw}}}/\Delta C}+\frac{\Delta C}{\Delta \Omega_{R}}\frac{2\pi }{2NT_{p,\min}}$$

The Rabi frequency depends mainly on the oscillating magnetic field perpendicular to the NV symmetry axis and the microwave power used. Slight differences in the angle between the $B_{\mathrm{mw}}$ component of the microwave and one of the [111] axes of different NV samples will lead to different Rabi frequencies. To determine the straight-line equation for a given resonance, it must be calibrated initially using at least two measured frequencies. Fast Fourier transformation (FFT) analysis is a widely used and well-understood method to decompose a signal into its frequency components. Therefore, a routine for calibration using the rfft function from the CMSIS library [25] is implemented on the microcontroller, which transforms the Rabi signal into the frequency domain and filters out the dominant frequency. The sampled signal is stored in an array and to minimize the DC component, the value of the last sample is subtracted from the entire array at the end of the measurement. The signal is only recorded for 30 values because, due to a small decay time constant $T_{2}^{\mathrm{rabi}}$, it does not contain any more significant information and would only drag out the measurement. Instead, zero padding is used to stretch the signal to 256 values before transforming it.

Zero padding does not improve the real frequency resolution, but the FFT output signal is interpolated and can be processed by a peak detection function. Figure 7a shows the calculated spectra for two different DAC values. The colored points are the resonance frequencies where we measured a shift of 2.197 MHz for a DAC difference of 1000. From these values follows the calculation of the required parameters with $\Delta C/\Delta \Omega_{R}=-4096/18\pi$ and $\left(b_{\Omega_{R}}-b_{C}\right)/\left(\Delta \sqrt{P_{\mathrm{mw}}}/\Delta C\right)=2200$, which the microcontroller uses for the application of Equation (Equation 8). Figure 7c shows the application of the equation for different pulse lengths of the size $NT_{p,\min}$ of the microwave in blue and represents the time of the measured π pulse. For this measurement, the respective Rabi oscillation was fitted to Equation (Equation 5), the result was derived and the first zero crossing was calculated. The ideal curve is shown as a black dashed line and shows that the measured π pulse duration deviates from the pulse length of the microwave. The reason for the deviation is the changing phase of the fundamental frequency, which leads to a standard deviation of $\sigma_{b}=42.77$ ns between the blue data points and the dashed line. The relationship between the phase position of the Rabi oscillation and the DAC output can be calculated directly from the FFT and is shown in Figure 7b in radians. The dashed regression line also indicates a linear relationship to the DAC value. The temporal deviation can be calculated from the phase using the relationship $t_{\mathrm{diff}}=\phi_{R}/2\pi f_{R}$, based on an initial phase of the Rabi frequency of π (180°). Combined with the linearity, $t_{\mathrm{diff}}$ is used in the microcontroller to correct the output value of the DAC in order to minimize the standard deviation between the π pulse and the length of the microwave pulse. First, the register value C is calculated according to Equation (Equation 8) and inserted into the linear equation of the phase position. The correction value is calculated according to the following: (9)$$t_{\mathrm{adj}}=\frac{\Phi_{R}}{2\pi \frac{1}{{\mathrm{NT}}_{p,\min}}}$$
and is added to the denominator of the original Equation (Equation 8) to calculate the adjusted register value. The red data points in Figure 7c are the same measurements as in blue, but with a correction of the phase shift. The resulting standard deviation is then reduced to $\sigma_{r}=7.53$ ns.

## 5. Conclusions and Outlook

In summary, we present a low-cost and portable measurement device for NV centers based on a microcontroller. The pulse sequences required for typical measurements, such as ODMR, Rabi oscillation, or Hahn echoes were recorded via a programmable pulse pattern, and the resulting fluorescence change of the diamond was measured with the ADC of the microcontroller. The linear dependence of the Rabi frequency on the output of the DAC was utilized to match the π pulses required for coherent control to the limited temporal resolution of the microcontroller’s GPIO pulses. The process yielded a standard deviation of $\sigma_{r}=7.53$ ns, which is better compared to similar microcontroller-based systems with 62.5 ns [5] but does not come close to the resolution of commercially available pulse generators with 2 ns [8]. FPGA-based setups also achieve minimal pulse widths in the single-digit nanosecond range [26,27]. However, the measurement electronics presented in this work still leave enough room for future improvements. We expect to further increase the accuracy by making the FFT resolution finer to better fit the linear equations of Equations (Equation 8) and (Equation 9). This can be achieved by using a less doped NV diamond, which results in a longer coherence time [17,28] and the Rabi oscillation can be sampled for a longer period of time. Additionally, a decrease in microwave power with increasing time was observed, which has a direct influence on the Rabi oscillation and requires further investigation for compensation. Overall, other parameters such as the temperature dependency of the microwave power of the unregulated microwave source can be determined in order to include them in the calculation of the DAC value. Despite the possibilities for improvement, the handheld measuring device shown here is the prototype that was produced with a total material cost of less than USD 300. There are exclusively commercially available electronic components on the circuit board, which makes machine assembly possible. The existing demonstrator can therefore be produced on an industrial scale and the costs reduced even further. The total power consumption is only 5.4 W and offers further savings potential by improving the RF technology. However, the device can already be operated with a battery pack, making it very easy to handle. Combined with the simple programming of the pulse sequences, which are required for coherent control, their use in educational settings will be promising, as students can be introduced to future technologies and learn about quantum-based sensor technology or quantum computing. The developed system also offers advantages outside of laboratory environments. By replacing the SLA-printed structure with a compact sensor head, as in Pogorzelski et al. [2], or by extending a fiber-coupled system, a variety of interesting applications arise for the commercial sector. With the four orientations of the NV centers, vector magnetometry for static or slowly changing magnetic fields can be realized using ODMR. If the crystal orientation is known, the parallel $B_{0}$-field component can be determined by splitting the resonances and the angle of incidence can be calculated back. Homrighausen et al. [29] have already demonstrated this in a fiber-based setup. The Hahn echo, in which the NV center becomes sensitive to alternating magnetic fields whose frequency corresponds to twice the pulse spacing τ, is suitable for measuring AC fields. This approach is investigated in more detail in [30]. The coherence time of the diamond used determines the measurable frequencies, which can be between the kHz and MHz range. Temperature measurement is also possible by using the $T_{1}$ time. Mrozek et al. [14] show a comparison of the $T_{1}$ times of different diamond samples as functions of temperature, showing a change in 1/$T_{1}$ by several orders of magnitude in a range from 100 to 500 Kelvin. The great advantage of NV centers over other magnetic sensors is the high local resolution that can be achieved. The diamond used has a size of 150 μm. Single NV centers are already available in the lower nanometer range and show how far the local resolution can be further increased. Very sensitive detectors, such as avalanche photodiodes (APDs), which are operated with a high bias voltage of several 100 V, are required to measure individual NV centers. Since such a high-voltage source is missing in the hardware, it is not suitable for the measurement of individual NV centers. The operation of APDs in Geiger mode for the detection of single photons also presents further technical hurdles. Active quenching is necessary for a high count rate in order to reduce the dead time of the APD. In addition, a circuit with a sufficiently high bandwidth is required to be able to count the current pulses. However, the extension to measure individual NV centers is highly interesting for future development.

## Figures and Tables

**Figure 1 sensors-24-03138-f001:**
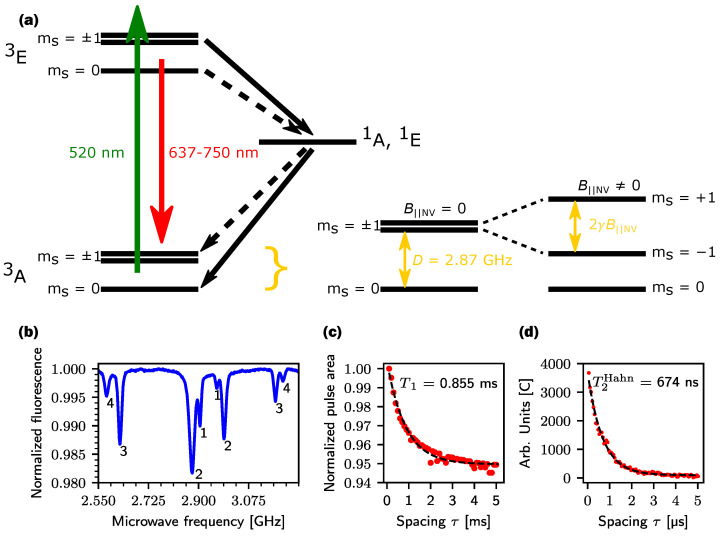
(**a**) Interrelationships of the individual energy levels in the NV center. Ground state ^3^A and excited state ^3^E are separated by 1.945 eV and divided into three $m_{S}$ states. The green arrow represents the optical pumping of electrons to the excited state, while the red arrow marks the light-emitting decay back to the ground state. The transition can also be non-radiative via the singlet states ^1^A, ^1^E back to the ground state and initialize the electron spins into the $m_{S}=0$ state. This path is shown by the black arrows. The dashed arrows represent lower-probability decay paths. (**b**) pulsed ODMR spectrum of the NV ensemble at a microwave power of 17 dBm and a pulse length of 1.2 μs. The four NV axes 1–4 correspond to the eight visible resonances at a static magnetic field $B_{0}$ perpendicular to the [111] orientation. (**c**) $T_{1}$ relaxation of the polarized electron spins from the $m_{S}=0$ state back into a Boltzmann distribution of the thermal equilibrium without using resonant microwaves. (**d**) $T_{2}$ relaxation of the second innermost $m_{S}=-1$ resonance using a Hahn echo sequence with a spin-flip time $T_{\pi }$ = 106.6 ns. The dashed lines in (**c**,**d**) correspond to a fit function of the form $Ae^{-t/T_{x}}+O$ with a decay time constant of $T_{1}=0.855$ ms and $T_{2}^{\mathrm{Hahn}}=674$ ns.

**Figure 2 sensors-24-03138-f002:**
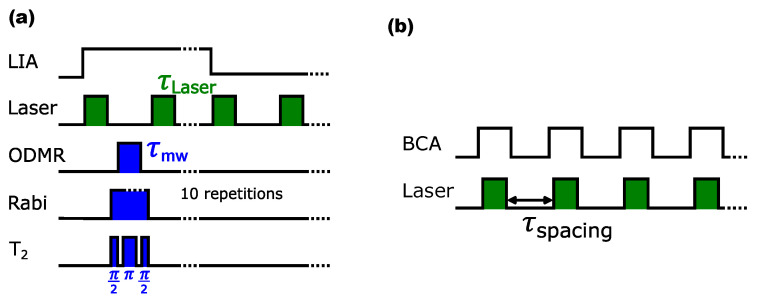
Measurement principle. (**a**) The pulse pattern of laser and microwave operations (ODMR, Rabi, T_2_) is repeated ten times in the first reference half-wave of the LIA. In the second reference half-wave, only the laser pattern is repeated ten times. At the LIA output, the difference is calculated from the fluorescence mean values of the reference half-waves. (**b**) The integration time window of the BCA only records signals while the laser is active. The $\tau_{\mathrm{spacing}}$ time is ignored. The mean value is then calculated from *N* integrals.

**Figure 3 sensors-24-03138-f003:**
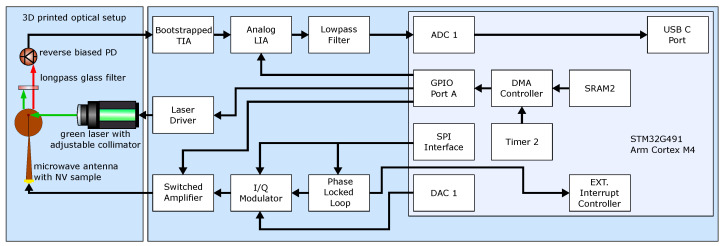
Schematic diagram of the measurement electronics and the optics as a setup in blue. The black arrows show the electrical paths between the individual modules, while red and green represent the fluorescence of the NV diamond and the excitation light. Light blue shows the peripherals used by the microcontroller.

**Figure 4 sensors-24-03138-f004:**
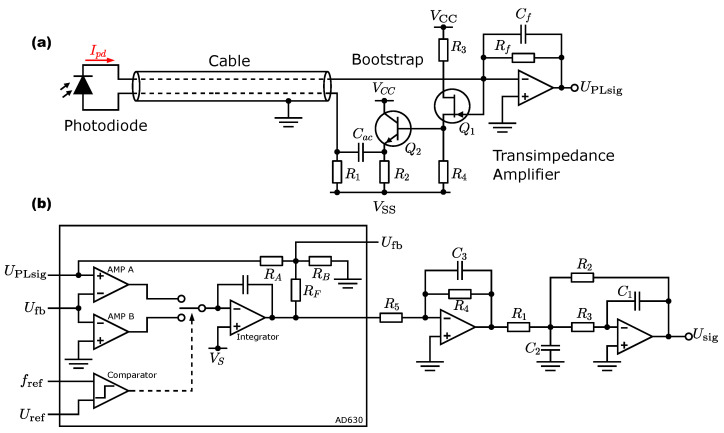
Electronics design for (**a**) bootstrapped photodetector with an additional transistor driver stage Q2. (**b**) The configuration of the AD630 IC as an analog LIA with a third-order active multiple feedback filter (MFB).

**Figure 5 sensors-24-03138-f005:**
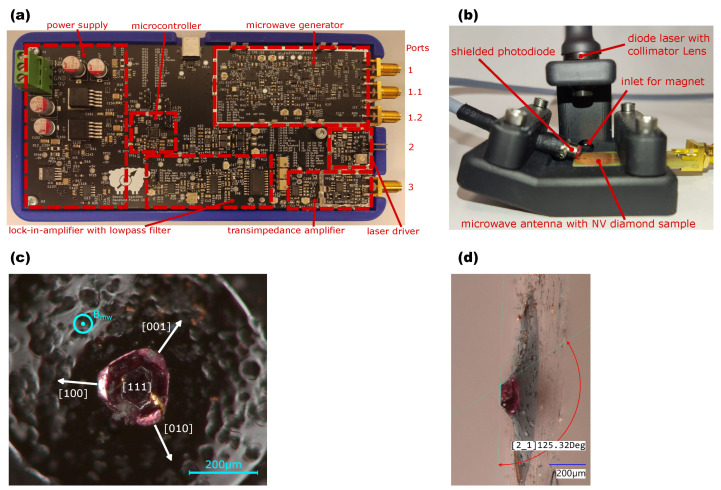
(**a**) Fully assembled demonstrator in an SLA-printed package. The dimensions are 82 mm × 167 mm. (**b**) Optical assembly, manufactured with a stereolithography (SLA) printer. The simple design uses a minimum of optical components and only requires an adjustable collimator lens for the laser diode as well as a filter for the photodiode. (**c**) Microscope image of the 200 μm NV diamond in the center hole of the antenna [23] at a front view. Crystal axes are drawn based on the geometry with the [111] orientation perpendicular to the antenna. (**d**) Side view with the measured angle between the (111) and (001) planes.

**Figure 6 sensors-24-03138-f006:**
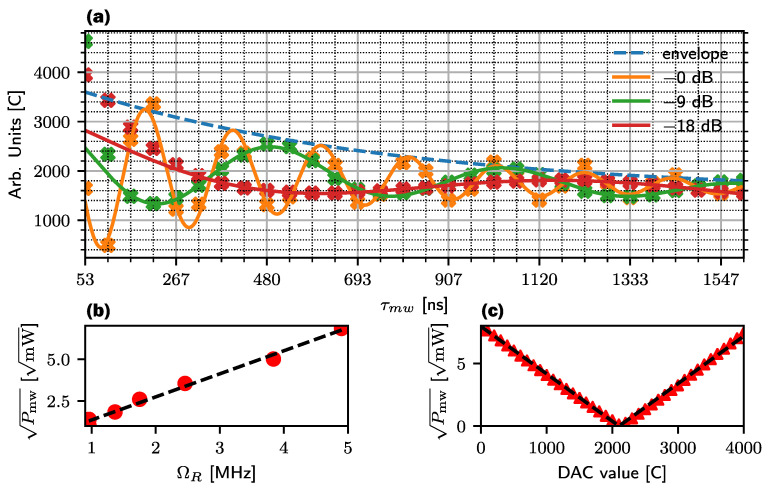
(**a**) Change of the Rabi oscillation of the second innermost $m_{S}=-1$ resonance when the microwave power is reduced to one-eighth step. The blue line refers to an interpolated and fitted exponential decay to the 0 dB curve. (**b**) Extracted Rabi frequency from the found fit parameters of Equation (Equation 5) versus the root of the power. The dependency can be described as a linear relationship. (**c**) Step increase in DAC value at the I input of the I/Q modulator at a fixed frequency of 2.87 GHz. The red measuring points have a distance of 100 and also show a linear relationship to the output power.

**Figure 7 sensors-24-03138-f007:**
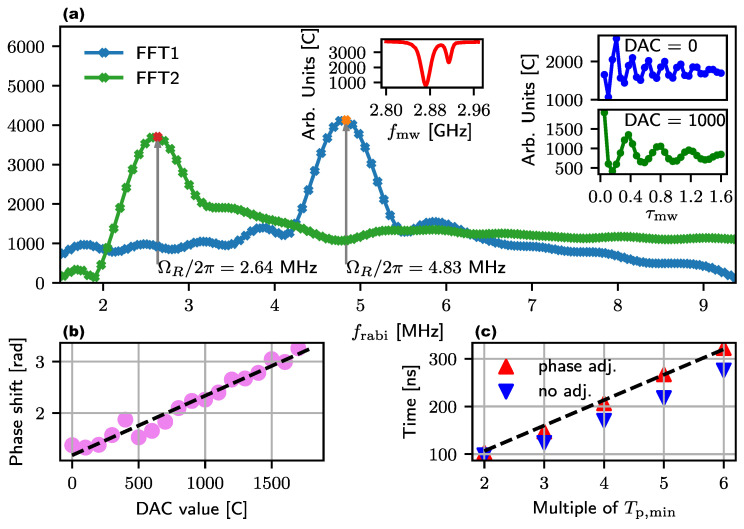
(**a**) FFT spectra calculated by the microcontroller correspond in color to the Rabi oscillation insets. The measured frequency in resonance at 2.872 GHz (red ODMR signal) shifts from 4.83 MHz (blue) to 2.64 MHz (green) for a DAC change of 1000. (**b**) Phase shift of the fundamental Rabi oscillation for decreasing microwave power through the DAC from an initial position of π. (**c**) Time to reach the $m_{S}=\pm 1$ state for a microwave pulse of length $NT_{p,\min}$ in blue using Equation (Equation 8). Extension of the equation by a compensation of the phase shift in red.

**Table 1 sensors-24-03138-t001:** Component calculation with filter coefficients and the corresponding values for a Butterworth characteristic. Index 1 for the first-order filter and index 2 for the second-order filter.

Coefficient	Butterworth	Formula
$a_{1}$	1	$\omega_{g}R_{4}C_{3}$
$a_{2}$	1	$\omega_{g}C_{1}\left(R_{2}+R_{3}+\frac{R_{2}R_{3}}{R_{1}}\right)$
$b_{2}$	1	$\omega_{g}^{2}C_{1}C_{2}R_{2}R_{3}$
$A_{1}$	–	$-\frac{R_{4}}{R_{5}}$
$A_{2}$	–	$-\frac{R_{2}}{R_{1}}$
$f_{g1}$	1 Hz	–
$f_{g2}$	1.272 Hz	–

## Data Availability

Data underlying the results presented in this paper are not publicly available at this time but may be obtained from the authors upon reasonable request.

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
