# Peer review of "Microcontroller-Optimized Measurement Electronics for Coherent Control Applications of NV Centers"

_sensors, 2024, doi:10.3390/s24103138_

Round 1

Reviewer 1 Report

Comments and Suggestions for Authors

The authors proposed a measurement elecctronic for NV centers in order to optimize microcontrollers. This is interesting topic and may important for designing novel NV sensors. I still has some questions as follows.

Line 66, why is the theoretical 30% contrast of the spin signals not achieved? What is the main reason affecting the constrast of the spin signals?

L141, "the decaying echo is caused by fluctuating magnetic fields". Does fluctuation of magnetic fields need some time to fit the system's steady state? Will it destroy the coherence of the system and further affect coherent control of spin system?

Line 199, The absorption of photons in the bandgap of the photodiode results in the shot noise by the random generation of electron-hole pairs around a mean value, while, thermal noise is coupled into the signal via the feedback resistor Rf. What is difference between two noise? How about their mechanism and influence on the signals?

Reviewer 2 Report

Comments and Suggestions for Authors

The manuscript presents an innovative approach to designing a low-cost and portable measurement device tailored for NV centers. The authors effectively utilize microcontroller technology to record pulse sequences essential for various measurements such as ODMR, Rabi oscillations, and Hahn echoes. The linear dependence of the Rabi frequency on the DAC output is cleverly exploited to adapt π pulses to the microcontroller's limited temporal resolution. But there are still fundamental questions to be clarified.

1The manuscript lacks clarity regarding the descriptions for figures 1b, 1c, and 1d. These three subfigures are not adequately explained within the text.

2The phrase "a visible optical contrast of luminescence of maximum 30%" requires clarification. It is unclear whether this refers to the maximum ODMR contrast being 30% or if it pertains to another aspect of the luminescence.

3Consider including a discussion on the scalability and potential applications of the proposed handheld setup beyond academic research, such as in industrial or commercial settings. This will broaden the paper's relevance and appeal to a wider audience.

Comments on the Quality of English Language

Moderate editing of English language required

Reviewer 3 Report

Comments and Suggestions for Authors

In their paper, Dennis Stiegekötter et al. reported measuring NV centers through low-cost and portable microcontroller-optimized measurement electronics. Although the measurement device can be further optimized, their discussion is thorough and convincing, and their results already showed the potential of such low-cost and portable devices for educational purposes and can promote the development of practical devices based on NV centers. Therefore, I would commend accepting this manuscript for publication after the authors address my comments below:

(1)     Since the gap between ms=-1 and =+1 is affected by a static magnetic field-B0, could the authors discuss how the value B0 affects the device performance, especially when B0 is obviously larger than 15 mT?

(2)     It is ideal to measure a single NV center for quantum devices. Can the authors provide some insights on whether the proposed device configuration can be further used for single NV centers?

Reviewer 4 Report

Comments and Suggestions for Authors

The manuscript "Microcontroller-Optimized Measurement Electronics for Coherent Control Applications of NV Centers" presents the results of development and testing the newly designed electronic blocks designated for measuring optically detected magnetic resonance (ODMR) parameters in nitrogen-vacancy (NV) centers in diamond. This topic is relevant and timely, although the focus on electronics might make the manuscript slightly too technical for the "Sensors" journal.

The authors demonstrate a good technical and physical level. They show a good agreement between the calculated and experimentally measured parameters of the designed circuit. I especially appreciate the thorough consideration of noise in the amplifier circuit (Section 3.2. Bootstrapped Transimpedance Amplifier). I have only a few minor questions and comments:

1. Figure 1b: Please number the resonances in the spectrum and indicate the frequency corresponding to the longitudinal zero-field splitting (ZFS). What is the reason for the asymmetry of the spectrum wings?

2. Lines 49-51: "The energetic transition from the ground state 3A to the excited state 3E can be achieved by optical pumping at a wavelength of 520 nm" – In reality, the range of pumping wavelengths is much wider, extending approximately from 500 to 630 nm.

3. Figure 2: Please provide a more detailed explanation in the caption.

4. Section 2.2. “Detection of Spin Signals”: It seems that the authors dedicate too much space to well-known concepts such as definitions of relaxation times and pulsed methods for their measurement. I believe this section could be shortened.

5. Line 223: "The bootstrap keeps the photodiode and the cable at a constant voltage level and counteracts a change in voltage" – This statement raises some doubts. Firstly, the feedback signal is not obtained directly from the photodiode, but from the output of the cable connected to the operational amplifier (OA) input – that is, from the point where the voltage is maintained at zero by the OA feedback. Secondly, only the AC part of the signal formed in this way is transmitted to the photodiode, while the DC voltage is determined by the voltage drop across resistor R1 due to the photocurrent. I would like to ask the authors to comment on this.

I believe that after addressing these minor comments, the manuscript can be published in Sensors.

Round 2

Reviewer 2 Report

Comments and Suggestions for Authors

None